# Sex-Dependent Protective Effect of Combined Application of Solubilized Ubiquinol and Selenium on Monocrotaline-Induced Pulmonary Hypertension in Wistar Rats

**DOI:** 10.3390/antiox11030549

**Published:** 2022-03-14

**Authors:** Tatyana Kuropatkina, Olga Pavlova, Mikhail Gulyaev, Yury Pirogov, Anastasiya Khutorova, Sergey Stvolinsky, Natalia Medvedeva, Oleg Medvedev

**Affiliations:** 1Faculty of Medicine, Lomonosov Moscow State University, Lomonosovsky Prospect 27-1, 119991 Moscow, Russia; ofleurp@mail.ru (O.P.); gulyaev@physics.msu.ru (M.G.); medvedev@fbm.msu.ru (O.M.); 2Faculty of Physics, Lomonosov Moscow State University, Leninskie Gory 1-2, 119991 Moscow, Russia; yupi937@gmail.com; 3Research Center of Neurology, Laboratory of Clinical and Experimental Neurochemistry, Volokolamskoye Shosse 80, 125367 Moscow, Russia; hutorova.anastasiya@mail.ru (A.K.); stvolinsky@neurology.ru (S.S.); 4Faculty of Biology, Lomonosov Moscow State University, 1-12 Leninskie Gory, 119234 Moscow, Russia; medvedeva@mail.bio.msu.ru; 5National Medical Research Center of Cardiology of the Ministry of Health of the Russian Federation, Laboratory of Experimental Pharmacology, 3rd Cherepkovskaya 15a, 121552 Moscow, Russia

**Keywords:** CoQ10, ubiquinol, selenium, ROS, pulmonary hypertension, endothelium dysfunction, pulmonary fibrosis, ^1^H MRI, ^19^F MRI

## Abstract

Ubiquinol exhibits anti-inflammatory and antioxidant properties. Selenium is a part of a number of antioxidant enzymes. The monocrotaline inducible model of pulmonary hypertension used in this study includes pathological links that may act as an application for the use of ubiquinol with high bioavailability and selenium metabolic products. On day 1, male and female rats were subcutaneously injected with a water-alcohol solution of monocrotaline or only water-alcohol solution. On days 7 and 14, some animals were intravenously injected with either ubiquinol’s vehicle or solubilized ubiquinol, or orally with selenium powder daily, starting from day 7, or received both ubiquinol + selenium. Magnetic resonance imaging of the lungs was performed on day 20. Hemodynamic parameters and morphometry were measured on day 22. An increased right ventricle systolic pressure in relation to control was demonstrated in all groups of animals of both sexes, except the group of males receiving the combination of ubiquinol + selenium. The relative mass of the right ventricle did not differ from the control in all groups of males and females receiving either ubiquinol alone or the combination. Magnetic resonance imaging revealed impaired perfusion in almost all animals examined, but pulmonary fibrosis developed in only half of the animals in the ubiquinol group. Intravenous administration of ubiquinol has a protective effect on monocrotaline-induced pulmonary hypertension development resulting in reduced right ventricle hypertrophy, and lung mass. Ubiquinol + selenium administration resulted in a less severe increase in the right ventricle systolic pressure in male rats but not in females 3 weeks after the start of the experiment. This sex-dependent effect was not observed in the influence of ubiquinol alone.

## 1. Introduction

Coenzyme Q10 (CoQ10) is an endogenous electron carrier in the mitochondrial respiratory chain of almost any cell in the body. In the process of cellular respiration, it binds and releases electrons and protons and is subsequently converted from ubiquinol to ubiquinone. It has been established that about 95% of CoQ10 in the human body exists as a reduced form, ubiquinol, which has a high redox potential and, as a consequence, more pronounced antioxidant properties [1].

According to the literature reports, three out of four patients with cardiovascular disease show low CoQ10 levels [2]. There are three suggested mechanisms of CoQ10 activity in cardiovascular disease treatment. First, CoQ10 as ubiquinol can effectively suppress free radical oxidation, i.e., exhibits antioxidant properties [2,3]. It is known that the products of oxidative stress and cytokines can result in cardiomyocyte hypertrophy [2,4]. Secondly, CoQ10 is essential for the energy demands of the heart: it is a major component in the electron transport chain required for adenosine triphosphate (ATP) production [5]. Thirdly, oxidative stress directly provokes the inflammatory process and cardiovascular disease, such as heart failure is closely associated with chronic inflammation involving increased circulating levels of cytokines and adhesion molecules [6].

Another important substance is the essential trace element selenium. A blood selenium concentration of at least 115–120 μg/L is necessary for health support [7]. In the human body, selenium is found in many of enzymes, most of which, similarly to CoQ10, act as electron donors in redox reactions. It has been found that a selenium deficit is directly related to the development of cardiovascular pathology [8,9]. The glutathione peroxidase family is also of special importance, participating in antiperoxide cell protection. Both selenium and glutathione peroxidase exhibit antioxidant and cardioprotective properties preventing myocardial hypertrophy and reducing its ischemia and necrosis in acute hypoxic states [10,11,12,13,14].

CoQ10 and selenium occurring in functional selenoproteins have similar properties. In addition, the selenium-dependent enzyme, cytosolic thioredoxin reductase, is the most efficient ubiquinone reductase that enhances the antioxidant effect of ubiquinol [15,16]. A five-year study performed by Swedish scientists found that oral administration of selenium + CoQ10 combination more than halved the risk of death from cardiovascular disease in elderly people. It was noted that the marker N-terminal pro-brain natriuretic peptide (NT-proBNP) level was reduced in the group taking the supplements, indicating a lower risk of heart failure development [17]. The follow-up 4-year observation demonstrated a high quality of life in these patients and decreased hospitalization time compared to the placebo group [18,19].

Pulmonary hypertension (PH) is defined as a chronic disease characterized by elevated resistance of pulmonary vessels as well as their remodeling resulting in increased blood pressure in the pulmonary circulation and right ventricle hypertrophy (RVH) [20]. In the chronic form of PH, the specific morphofunctional changes have a predominantly inflammatory nature and include the processes of the apoptosis-resistant proliferation of endothelial and smooth muscle cells of the pulmonary artery. The inflammatory process in endothelial cells increases endothelial layer permeability and reduces NO-synthase activity causing endothelial dysfunction. In addition, the smooth muscle sensitivity to vasoconstrictors increases, which leads to the narrowing of arterioles lumen and right ventricle (RV) compensatory hypertrophy [20,21,22]. Therefore, oxidative stress and inflammation in endotheliocytes are among the important factors of PH pathogenesis and are potential targets for ubiquinol and selenium action.

Based on the foregoing, the aim of this work aimed to evaluate the effect of parenterally administered solubilized ubiquinol, orally administered organic selenium powder and their combination on the main aspects of the 3-week monocrotaline-induced PH development in male and female Wistar rats, as well as to assess the antioxidant status by measuring the selenium-dependent enzyme glutathione peroxidase action.

We also used magnetic resonance imaging (MRI) for lung status monitoring [23]. Beyond the structural lung assessment, which was carried out by proton MRI, we estimated the ventilation of lungs using the MRI method based on the detection of fluorine-19 nuclei (^19^F). This technique is novel and promising [24].

## 2. Materials and Methods

### 2.1. Animals and Experimental Design

The experiments were carried out on 2-month-old male and female Wistar rats (180–220 g). All the manipulations with the animals were carried out according to the Council Directive 86/609/EEC principles. The animals were obtained from the vivarium of the Research Institute of General Pathology and Pathophysiology (Moscow, Russia). The rats were kept under 12 h daylight conditions with free access to water and food, humidity and temperature control was also performed. The adaptation period after transportation was at least 7 days. Later the rats were weighed, their systolic blood pressure was twice measured by tail-cuff plethysmographic technique. In the experiment carried out on female rats with preserved ovaries, it was important to take into consideration the hormonal cycle in order to standardize the manipulation.

Monocrotaline-induced (MCT) PH model was chosen as the most convenient, technically simple and reproducible. Its pathogenesis includes all the special links: oxidative damage and inflammatory response, endothelial dysfunction, vascular remodeling, smooth muscle cell proliferation of pulmonary vessels and cardiomyocytes which can be potentially influenced by the substances under study [25].

On day 1, the four groups of animals received a single subcutaneous injection of monocrotaline (MCT) (60 mg/kg in 60% ethyl alcohol) (Sigma Aldrich, Darmstadty, Germany). The control group was administered subcutaneously only with a solvent for MCT (60% ethyl alcohol), i.e., it was the control to the MCT action.

On day 7, the Wistar rats were randomized into groups (*n* = 10♂/12♀ in each group): the MCT-Ubiquinol group receiving into the tail vein 1% of solubilized ubiquinol (RU2635993-C1) at a dose of 30 mg/kg (composition (1 mL) ubiquinol 10 mg, macrogol glycerol ricyl oleate 80 mg, polysorbate 40 mg, ascorbic acid 1 mg, sodium EDTA 0.5 mg, sodium chloride 0.9% ad 1 mL); the MCT-Vehicle group receiving vehicle for the ubiquinol substance (the same composition, but without ubiquinol, i.e., it was the control group to ubiquinol administration); the MCT-Se group receiving intragastric administration of selenium at a dose of 10 µg/kg (100 µg, Solgar, Leonia, NJ, USA) and the MCT-Se-Ubiquinol group receiving the combination of ubiquinol and selenium powder. The control group of healthy rats was also injected with a vehicle without ubiquinol. The intravenous administration of substances was repeated on day 14, the intragastric administration of selenium powder was performed daily over a period of 14 days. Since the MCT-Ubiquinol and MCT Se-Ubiquinol groups of male rats did not reveal any significant increase in lung mass, the subsequent experiment on female rats involved MRI of lungs on day 20 to detect fibrotic and unventilated regions and check the assumption of the ubiquinol influence. On day 22, the hemodynamic parameters (mean arterial pressure, heart rate, right ventricle systolic pressure were estimated through a catheter directly, venous blood was preserved; after that the animals were sacrificed by cerebral dislocation method, the morphometric study of the internal organs was performed and part of the liver was preserved. The scheme of time and manipulations is shown in Figure 1.

A wide range of oral doses of selenium is used in experiments on animals; however, if the physiological requirement is exceeded, selenium can become toxic. Since small quantities (calculated for an average rat weight of 200–300 g) are very easy to overdose, a dosage of 10 µg/kg was chosen [26,27].

### 2.2. Measurement of Systolic Pressure in the Tail Artery

Indirect measurement of systolic arterial pressure (SAP) was performed by plethysmographic method every 7 days throughout the experiment. SAP registration was carried out using the LGraph software (version 1, L-Card, Moscow, Russia). At each time point, 5–7 such measurements were taken and then the mean SAP was calculated for each rat.

### 2.3. Cycle Phase Determination

We started the manipulations at least 9 days before and continued throughout the experiment to create a calendar for each animal. A smear was taken by injecting a small amount of isotonic sodium chloride solution into the vagina followed by carefully collecting the solution back into the syringe; afterwards it was placed on a slide and the phase of the cycle was determined according to the cells’ shape using a binocular microscope. MCT injection and acute experiment were performed in the diestrus phase since estradiol potentially protective and vasodilatory effects are minimal.

### 2.4. MRI Study

The experiments were performed on 7T MR scanner Bruker BioSpec 70/30 USR (Bruker BioSpin, Ettlingen, Germany). The system is equipped with a horizontal superconducting magnet with a bore diameter of 30 cm, 1 kW radiofrequency (RF) transmitter and gradient system with 100 mT/m power and 0.2 ms rise time.

As a transceiver, a volume resonator (a birdcage) with 7 cm internal diameter was applied. The tuning range of the birdcage was sufficient to operate at the frequency of hydrogen (~300 MHz) and fluorine nuclei (~283 MHz). To increase the detected ^19^F MR signal we also used a wireless radiofrequency (RF) coil with 4 cm diameter—multi-turn multi-gap transmission line resonator (MTMG-TLR) that constitutes two concentric conductive circuits at the opposite sides of a dielectric sheet [28]. The MTMG-TLR was applied as inductively coupled with the birdcage.

For in vivo MRI, rats were anesthetized with i/p injection of 300 mg/kg (12%) chloral hydrate, and then intubated using BioLite Small Animal Intubation System (Braintree Scientific Inc., Braintree, MA, USA) with the help of i/v catheters (18 G–1.3 mm). The animals were placed on their backs in small animal MRI bed and the MTMG-TLR was placed on their thorax. The artificial lung ventilation (ALV) machine—7025 Rodent Ventilator (Ugo Basile S.R.L., Comerio, Italy) was applied for pumping lungs with gas mixture consisting of 20% of oxygen and 80% of fluorinated gas octafluorocyclobutane (OFCB, C_4_F_8_) [28].

^1^H MR images were obtained using 3D UTE pulse sequence with the following scan parameters: field of view = 7 × 7 × 7 cm^3^, matrix size = 156 × 156 × 156 (0.45 mm isotropic resolution), TR (repetition time) = 8 ms, TE (echo time) = 0.06 ms, flip angle = 30° (8 µs block pulse), number of aver (echoages = 2, bandwidth = 25 kHz, acquisition time = 20 min 20 s.

^19^F MR images were acquired using the same 3D UTE pulse sequence and the scan parameters were as follows: field of view = 10 × 10 × 10 cm^3^, matrix size = 64 × 64 × 64 (1.6 mm isotropic resolution), TR (repetition time) = 8 ms, TE (echo time) = 0.06 ms, flip angle = 55° (0.1 ms block pulse), bandwidth = 25 kHz, number of averages = 3, acquisition time = 5 min 7 s.

Reconstruction of the MR images was performed using proprietary software (ParaVision v.5.1, Bruker Biospin, Ettingen, Germany). ImageJ software (v.1.51j8, NIH, Bethesda, MD, USA) [29] was used for the subsequent image processing and analysis procedures, reslicing, etc.

### 2.5. Registration of Hemodynamic Parameters In Vivo

The animals were anesthetized with urethane (aqueous solution, 1.2 g/kg, 0.6 g/mL) intraperitoneally. Mean arterial pressure (MAP) and right ventricle systolic pressure (RVSP) were evaluated directly using the Statham blood pressure transducer (Statham Instrument Inc., Los Angeles, CA, USA), an operational amplifier and the L-Card E14–140 multichannel analog-to-digital converter (version 1, L-Card, Moscow, Russia). For that purpose, the anesthetized rats were injected with the PE10 catheter into the femoral artery (MAP, heart rate (HR)) and the PE 50 catheter (Medsil, Moscow, Russia) was injected through the right jugular vein into the RV. The RVSP rate was used to evaluate the severity of pulmonary hypertension.

### 2.6. Morphometric Measurement

A morphological study of the myocardium was carried out after the hemodynamic indices’ registration. After euthanasia the heart was taken out and washed in a saline solution; the atrium was cut out, the left ventricle was separated from the right ventricle and the interventricular septum. The assessment of the RV hypertrophy degree was estimated relative to the sum of the left ventricular (LV) and interventricular septum (S) mass (RV/(LV + S)) in conventional units. The lung mass was also measured.

### 2.7. Determination of Glutathione Peroxidase Activity in Venous Blood Hemolysates

The collected venous blood samples were centrifuged (3 min at 3000 rpm), plasma was collected and stored for further analysis, 20 mL of erythrocyte suspension was carefully collected from the bottom of a vial and an aliquot was mixed with 380 mL of deionized water; after intensive shaking the resulting hemolysate was frozen and stored at −80 °C.

The glutathione peroxidase (GPx) activity in venous blood hemolysate was determined by the method [30] in a coupled system with glutathione reductase (GR) by the rate of nicotinamide adenine dinucleotide phosphate (NADPH) oxidation that is used for the reduction of oxidized glutathione (GSSG). Hydrogen peroxide was used as a glutathione peroxidase substrate.
GPx
H_2_O_2_ + 2GSH → 2H_2_O + GSSG
GR
GSSG + 2NADPH → 2GSH + 2NADP^+.^

Measurements were carried out on an Ultrospec 3300 UV/Visible spectrophotometer (Amersham Biosciences, Upsalaa, Sweden). A measure of 50 μL of hemolysate was added to the reaction medium with a total volume of 2.45 mL containing 0.1 M K,Na-phosphate buffer pH 7.0, 1 mM EDTA, 1 mM sodium azide, 1 mM reduced glutathione (GSH), 0.2 mM reduced NADPH and 2 U of glutathione reductase. The cuvette was placed in a spectrophotometer. The reaction was started by adding 5 µL 0.25 mM H_2_O_2_ prepared with ethanol, quickly mixed and the decrease in optical density was measured at λ = 340 nm during 1 min at 30 °C. The activity of the GPx was calculated using the following formula
$$A=\frac{\Delta D\times V_{mixture}\times 10^{9}\times K\;}{\varepsilon \times l\times T\times V_{hem}\times C_{protein}}\;\mathrm{nmol}/\min/\mathrm{mg}\;\mathrm{of}\;\mathrm{protein},\;$$
where:

Δ*D* (o.u)—optical density change of NADPH in the reaction mixture

*V_mixture_* (L)—volume of the reaction mixture (0.0025 L)

10^9^—conversion to nmol of NADPH

*K*—dilution factor of venous blood sample with deionized water (20, *V*/*V*)

ε—the extinction coefficient of NADPH (6620 × mol^−1^ cm^−1^)

*l* (cm)—optical path length (1 cm)

*T* (min)—incubation period (1 min)

*V_hem_* (L)—volume of added hemolysate (0.00005 L)

*C_protein_* (mg/mL)—concentration of total protein in hemolysate samples

The activity of GPx was expressed in oxidized NADP^+^ nmol/min/mg of protein.

Concentration of total protein in hemolysates was determined using DC Protein Assay Kit II #5000112 (Bio Rad, Hercules, CA, USA); the results were calculated using microplate reader Agilent BioTek Synergy H1 Multimode Reader (BioTek, Santa Clara, CA, USA); all procedures were performed in accordance with the protocol proposed by the manufacturer of the mentioned kit.

### 2.8. Method for Ubiquinol Determination in Plasma and Liver of Rats by High-Performance Gas-Liquid Chromatograph (HLPC)

After decapitation, the liver of the rats was removed and washed in saline solution, then dried with a paper napkin; a lobule was cut off, placed in Eppendorf tube, frozen and stored at −80 °C.

During chromatography run, the sample preparation was carried out according to the validated method [20]. Liver tissue was ground in a homogenizer and mixed with distilled water at the ratio of 1:4 m/V.

To extract analyte from the liver tissue, 220 μL of ethanol and 550 μL of n-hexane were added to 100 μL of plasma/homogenate. The mixture was thoroughly shaken for 10 min, centrifuged at 3000 rpm for 3 min and the top layer of n-hexane was collected. Then 550 μL of n-hexane was added once again and the extract collection was repeated. The pooled extract was evaporated and dissolved in ethanol. Quantitation was carried out by the HPLC method involving electrochemical detection using Environmental Sciences Associate Inc software (Chelmsford, MA, USA) equipment: model 580 pump and “Coulochem II” electrochemical detector in isocratic mode on the Luna 150 × 4.6 column with C18 sorbent (5 μm) at the eluent flow rate of 1.3 mL/min. The mobile phase is 0.3% of NaCl in the mixture of ethanol-methanol-7% HClO4 (970:20:10). Electrochemical detection was carried out in the oxidizing mode using an analytical cell (model 5011) with the voltage of −50 mV and +350 mV applied for the two pairs of electrodes, correspondingly. Chromatographic data registration and processing were carried out by using of the Environmental Sciences Associate Inc software (Chelmsford, MA, USA). The extract was analyzed both before and after the complete recovery of ubiquinol (by adding the solution of sodium tetrahydroborate in ethanol). The ubiquinol level registered before the reconstitution corresponded to that of the native, unoxidized substance. Adding the reducing agent converted the oxidized form into the reduced one and made it possible to determine the total CoQ10 concentration.

### 2.9. Bioethical Approvement

The research protocol was approved by the Bioethics commission of Moscow State University, Institute of Biology (Protocol 113-G, 19 June 2020).

### 2.10. Statistical Analysis

Statistical analysis of the results and randomization of the groups was performed using GraphPadPrism 8 (GraphPad, San Diego, CA, USA). Normality of distribution was checked using the Shapiro-Wilk test. One-factor analysis of variance (one-way ANOVA) was used to compare the mean values of a single index in more than two groups. Two-factor analysis of variance (two-way ANOVA) was used to establish the simultaneous effects of group and duration of exposure, as well as the interaction between these factors. The Kruskal-Wallis test was used to analyze the ranking data. Exclusion of statistical outliers was performed using the robust regression and outlier removal (ROUT) criterion with Q not exceeding 1%. Differences were considered statistically significant at *p* < 0.05. All data are presented as mean ± standard deviation (Mean ± SD). 

## 3. Results

In the process of the research, we studied the ubiquinol level in the experimental animals after double intravenous administration of 1% ubiquinol at a dose of 30 mg/kg. The object of the study was the liver tissue collected on day 22 in an acute experiment. Consequently, it was found that the ubiquinol level in the MCT-Ubiquinol and MCT-Se-Ubiquinol groups was more than 40 times higher than its average content in the other groups (*p* < 0.0001) (Figure 2).

Comparing the glutathione peroxidase action in blood hemolysate (Table 1) collected at the endpoint of the experiment, it appeared that its highest rate in male rats was shown in the MCT-Vehicle group and exceeded those in the control groups (*p* < 0.05) of MCT-Ubiquinol and MCT-Se (*p* < 0.01). The groups of females did not show such a difference. However, all the female groups except the MCT-Vehicle, showed a higher rate in this parameter in comparison with the male rats. (*p* < 0.01) (Table 1).

Systemic blood pressure and heart rate measurements did not reveal any statistically significant difference between the MCT-PH group compared with the control normotensive group of animals. The MCT-PH group demonstrated the MAP at the rate of 93 ± 13 and 91 ± 8 mm Hg, HR of 377 ± 28 and 356 ± 36 bpm in male and female rats, respectively; the control group showed on average 95 ± 10 mm Hg and 368 ± 35 bpm. Based on the data received, it can be concluded that the development of the 3-week MCT-induced PH model is not followed by systemic hypertension.

RVSP registration displayed that all the groups with MCT administration, except the MCT-Se-Ubiquinol male group (Figure 3A), exhibited its mean rate significantly higher in comparison with the groups of healthy animals (*p* < 0.05) which confirmed PH development (Figure 3). There was no significant difference in RVSP between the male and female experimental groups (Figure 3).

A significant increase of the right ventricular mass, i.e., its hypertrophy, in comparison with the control, was registered in the MCT-Vehicle and MCT-Se groups of both sexes (*p* < 0.05) (Table 2), suggesting PH development according to this marker in these experimental groups. The MCT-groups receiving intravenous injections of only ubiquinol or its combination with Se, displayed no differences in the right ventricular mass versus the control; i.e., these groups developed PH to a lesser extent. Taking into account the data on RVSP changes, it can be concluded that the MCT-Se-Ubiquinol group did not develop significant MCT-PH.

The lung mass in the MCT-Ubiquinol and MCT-Se-Ubiquinol groups did not notably differ compared with the control; the lung mass of other MCT groups, by contrast, was significantly higher (*p* < 0.05) (Figure 4). In females only, the MCT-Ubiquinol group did not exhibit any difference according to this indicator. No difference was found between male and female rats (Figure 4).

Since the effect of ubiquinol on lung mass firstly was noted in the experiment on males, which was before the experiment on females, an MRI study was performed in females two days before the acute experiment. The MRIs of two rats are shown in Figure 5. Fragment A represents a healthy rat, and fragment B, a rat with fibrotic changes in the lungs. In ^1^H MR images (upper rows of Figure 5), fibrosis looks like white (light) spots in the lungs. Fibrotic tissue prevents gas ventilation, which is visualized as signal lost in ^19^F MR images (middle rows).

Table 3 shows the data on the number of animals in each of the groups, which were diagnosed with lung disturbances according to ^1^H and/or ^19^F MRI. 

Almost all studied animals with PH revealed structural and/or ventilation disturbances according to the ^1^H and ^19^F MRIs performed. Pulmonary fibrosis in the MCT-Ubiquinol group was significantly less frequent than in the MCT-Vehicle, MCT-Se, and MCT-Se-Ubiquinol groups of female rats.

## 4. Results and Discussion

Being an essential component of the mitochondrial respiratory chain, coenzyme CoQ10 is involved in providing the bioenergetic processes in cells. In addition, CoQ10 acts as an important fat-soluble antioxidant and is characterized by an anti-inflammatory effect which explains its protective properties in various cardiovascular diseases (CVD) [1,31,32]. The previous research of the CoQ10 effect on MCT PH development in rats demonstrated that the two intravenous injections of ubiquinol contributed to a decrease of right ventricular (RV) hypertrophy in male rats with 3-week MCT-induced pulmonary hypertension [33]. At the same time, there was increased endothelium-dependent dilatation which still persisted when the viewing period lasted up to 4 weeks, which may have been one of the factors mediating the RV hypertrophy decrease [34]. Pulmonary hypertension development in experimental models is characterized by two main features: an increased RVSP and its hypertrophy. The previous research work did not reveal any ubiquinol influence on the RVSP value. The current study investigated the co-use of ubiquinol and selenium (Se), another antioxidant. As mentioned above selenium was chosen since several studies report that oral administration of a selenium and CoQ10 combination more than halved the risk of death from cardiovascular disease in elderly people. It is remarkable that the group taking supplements displayed a significantly reduced NT-proBNP level, suggesting a lower risk of heart failure development [17]. Since pulmonary hypertension has a pronounced gender dependence [35], the current study was performed involving both male and female rats.

It should be noted that oral CoQ10 administration is characterized by rather low bioavailability (0.3–3%) [36]. In this connection we used a solubilized pharmaceutical form of ubiquinol for parenteral use providing instant and continuous high tissue concentrations [36,37]. It was found that the ubiquinol level in liver tissue taken on day 22 of the experiment in the MCT-Ubiquinol and MCT-Se-Ubiquinol groups exceeded its average level in the other groups by 40 times, which is evidence of the high bioavailability of the chosen pharmaceutical form. The effectiveness of selenium administration (daily oral administration at a dose of 10 μg/kg) was determined by the activity of glutathione peroxidase (GTP), a Se-dependent enzyme showing strong antiperoxide defense activity [9,38,39,40,41]. The level of extra Se and methionine supply received in the body determines the proper maintenance of glutathione peroxidase and other selenium-containing enzyme rates [38]. Scientific findings reveal a sex-dependent difference in GTP activity. Female rats in all the experimental groups exhibited statistically significantly higher enzyme activity compared to that of males, which corresponds to the scientific data [40,41,42]. However, no difference in enzyme activity in female groups was detected, while the MCT-ubiquinol and MCT-Se male groups exhibited statistically significantly lower enzyme activity than that of the MCT male group. The obtained results do not correspond to the results of other studies. The probable reason may be insufficient quantity of selected animals coupled with the fact that the GTP level was not measured initially to get a more uniform randomization. Increased GTP activity in female rats did not affect the degree of MCT PH compared with male rats, no difference was found in the MCT-Vehicle group either. The data obtained indicate that changes in GTP activity ranging from 6.611 ± 0.593 to 10.880 ± 1.792 μmol NADPH/min × V sample have no impact on the degree of this type of PH. 

Analyzing RVSP changes in female rats with MCT hypertension showed that ubiquinol intravenous administration, oral administration of Se or their combination do not affect this marker of MCT hypertension development. In male rats, the RVSP value in the MCT hypertension group receiving the ubiquinol and selenium combination did not statistically significantly differ from the RVSP of the control group. RV relative weight in the same group was similar to that of the control. These facts suggest no hypertension development in this group, i.e., the co-use of ubiquinol and Se result in a protective effect and pronounced symptom levelling. It can be assumed that the mechanism of the observed effect is associated with CoQ10 ability to reduce endothelial dysfunction by stimulating NO synthase synthesis resulting in higher NO concentration and increased expansion of pulmonary vessels [43,44,45]. Owing to its position on the inner mitochondrial membrane, CoQ10 can able to increase both endothelial and mitochondrial NO synthase form (mNOS) activity, providing an optimal non-toxic level of endothelium-dependent expansion factor NO [6,43]. Acting as an antioxidant, CoQ10 is also capable of preventing NO oxidation to peroxynitrite and prolonging a vasodilatation effect. These findings are supported by the results obtained by Belardinelli et al., 2006, in patients with chronic heart failure [45] and Tiano et al., 2007, in patients with ischemic heart disease using oral CoQ_10_ form [46]. Kozaeva et al., 2017, has also demonstrated vasodilation of isolated aortic rings of healthy rats after incubation but with the studied form of ubiquinol [47]. In addition, previously we spotted an increase in the endothelium-dependent expansion of isolated segments of pulmonary artery against the background of intravenous administration of ubiquinol in the experiment on the MCT-induced PH model in male rats [33].

However, the effect of ubiquinol occurred only when it was coupled with selenium. It is known that selenium exhibits antioxidant properties suppressing signaling pathways and the adhesion molecules expression. In vivo and ex vivo experiments demonstrated a decreased expression of adhesion molecule-1, vascular cell adhesion molecule-1 and P-selectin being in charge of inflammatory response in endothelial tissue as well as its destruction [48]. In addition, selenium mediates functional selenoprotein synthesis, such as glutathione peroxidases and thioredoxin reductases, which belong to the first line of antiperoxide defense preventing the ROS damaging effect and endothelial cell apoptosis [9,10]. In the experiments described by Mo-Li Zhu et al., 2021, a significant decrease in RVSP and mean pressure in the pulmonary artery was noted in mice with MCT-PH on a background of organic selenium and nano-selenium supplement administration at a dose of 30 mg/kg/day in comparison with the MCT control. Nevertheless, in the course of our experiment, the use of oral selenium administration failed to have a similar effect, possibly owing to a lower dose and coarsely dispersed selenocysteine powder [49].

The gender difference in the obtained results can be connected with the estrogen cycle in female rats. Estrogen status alters tissue distribution and metabolism of selenium. Blood selenium and GPx activity were positively correlated with estrogen concentrations during the rat estrous cycle [50] and the human menstrual cycle [51]. We assume that the estrogen status affects tissue distribution of selenium by modulating selenoprotein P, as this protein plays a central role in selenium transport [52]. The duration of the estrous cycle in Wistar rats is 5–6 days and proestrus lasts 6–7 h [51], so the high content of estradiol counts and the increase in the consumption of selenium tissues were for only 5–6% of the duration of the cycle and probably had no effect.

The second marker of MCT hypertension development, RV hypertrophy degree, was comparable in both female and male rats [25,33,49]. The mean value of relative RV weight of rats of both sexes in the MCT-ubiquinol and MCT-Se-ubiquinol groups did not differ from that of the control animals, i.e., the animals in these groups did not develop hypertension according to this marker, in contrast, the MCT-control and MCT-Se groups exhibited a significantly pronounced hypertrophy.

Therefore, the results obtained indicate that the protective effect of ubiquinol + selenium administration on the RV hypertrophy degree in MCT-induced PH is associated mainly with the ubiquinol action. There is data evidencing that ubiquinol is able to prevent fibrotic tissue degeneration by participating in NF-kB/TGF-β1/MMP-9 pathway suppression [53]. Moreover, it is capable of decreasing the endothelin-1 (ET-1) level, which is one of the factors in the PH pathogenesis provoking cardiomyocyte proliferation [2,54,55].

However, there is probably a mechanism that selenium contributes to the ubiquinol action. It is associated with its inclusion in glutathione peroxidase and thioredoxin reductase [9,10] as well as other antioxidant enzyme repairs including superoxide dismutase [9,10]. In their study, Zamani Moghaddam et al., (2017) evaluated the effect of organic selenium and nanoselenium added to feed mixtures at a dose of 0.3 mg/kg. After 5 weeks of follow-up, it was concluded that the nanoselenium supplement significantly reduced right ventricular hypertrophy (RV/heart mass), reducing PH symptoms. The authors suggest that the mechanisms underlying this action involve decreased liver lipid peroxidation, immunomodulation, and enhancement in intestinal villus morphology [56]. The above-mentioned experiment performed by Mo-Li Zhu et al., (2021) demonstrated that organic selenium supplementation resulted in a significant hypertrophy decrease in mice with 3-week MCT-PH, but the authors gave no arguments explaining this effect [49].

Pulmonary hypertension is followed by fibrous (scar) tissue in the lungs along with RV hypertrophy resulting in impaired respiratory function. Cell damage leads to increased proinflammatory cytokine synthesis such as interleukin-1 (IL-1), IL-6 and tumor necrosis factor-α (TNF-α) [57,58,59], as well as one of the leading fibrogenic factors, connective tissue growth factor (CTGF) [60], chargeable for fibroblast proliferation. Structural changes inhibit normal pulmonary ventilation and perfusion resulting in frequent and shallow breathing, appetite and weight loss in animals [61]. Since the MCT-Ubiquinol and MCT-Se-Ubiquinol groups of male rats did not reveal any significant increase in lung mass, the subsequent experiment on female rats involved an MRI of lungs in order to detect fibrotic and unventilated regions to check the assumption of the ubiquinol influence. Therefore, it was found that fibrosis and impaired ventilation in the hypertensive control groups, i.e., MCT-Vehicle and MCT-Se groups, were more common than in the groups receiving ubiquinol detecting only impaired ventilation. This effect is probably mediated through the oxidative stress and inflammation suppression in MCT affected areas [62], as well as the mammalian target of rapamycin (mTOR) signaling pathway inhibition and increased autophagy marker expression as mentioned in the studies of Mohamed et al. [53]. Moreover, CoQ10 can directly inhibit the CTGF synthesis preventing cell proliferation, migration and adhesion in lung tissue [2,63], which probably explains the decreased lung mass both in the groups of male and female rats receiving only ubiquinol and in the male group receiving the combination of the two substances. The selenium supplement probably neutralized the ubiquinol action in the female group. No literary sources giving an idea of such specific activity in lungs, still, it may be associated with increased selenoprotein P expression in PH which is of prognostic significance [64]. However, it is still not entirely clear whether exogenous selenium supplementation provokes worsening of pathology and whether there is any sex dependence in selenium action.

## 5. Conclusions

Intravenous administration of ubiquinol at a dose of 30 mg/kg had a protective effect on MCT-induced PH development resulting in reduced RV hypertrophy, decreased lung mass and their fibrotic lesions both in male and female rats. Three weeks after the beginning of the experiment ubiquinol + selenium administration resulted in a slower increase in RVSP in male rats but not in females, which was different in the case of ubiquinol action. A daily intragastric selenium administration at a dose of 10 μg/kg was insufficient and may have been influenced by the estradiol cycle, that is why it probably did not lead to a significantly increased glutathione peroxidase activity, therefore demonstrating no pronounced effect on the main diagnostic features.

## Figures and Tables

**Figure 1 antioxidants-11-00549-f001:**
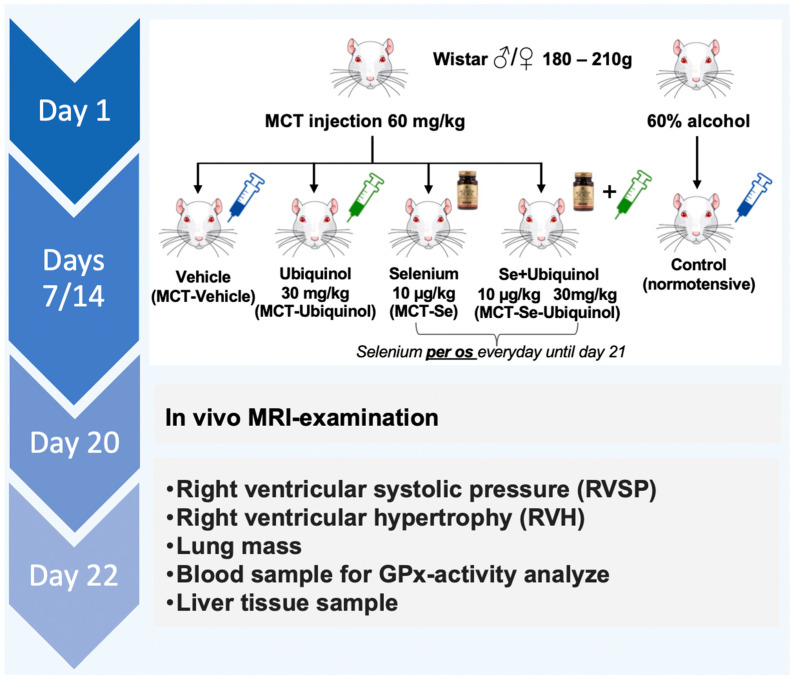
Scheme of the experiments. MCT: monocrotaline; MRI: magnetic resonance imaging.

**Figure 2 antioxidants-11-00549-f002:**
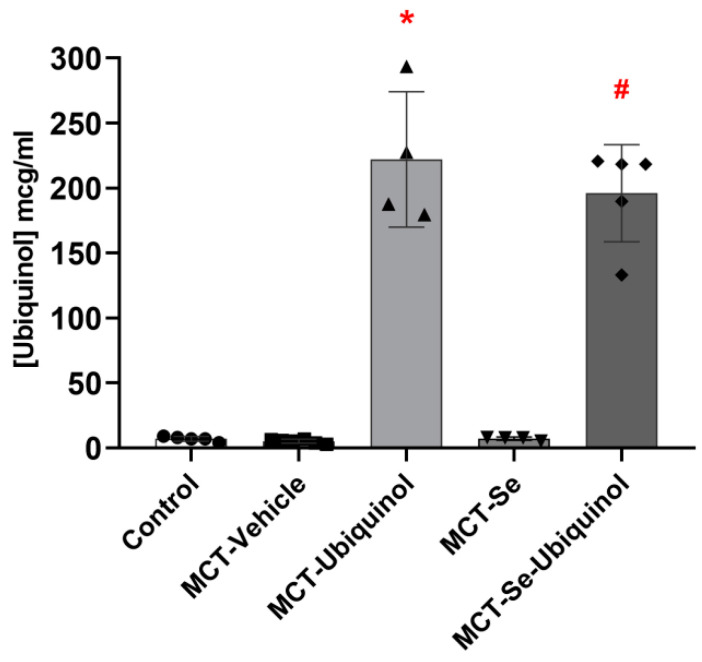
Ubiquinol content in the liver of male rats on day 21 (mean ± SD). * MCT-Ubiquinol, # MCT-Se-Ubiquinol vs. Control, MCT-Vehicle, MCT-Se, *p* < 0.0001, one-way ANOVA.

**Figure 3 antioxidants-11-00549-f003:**
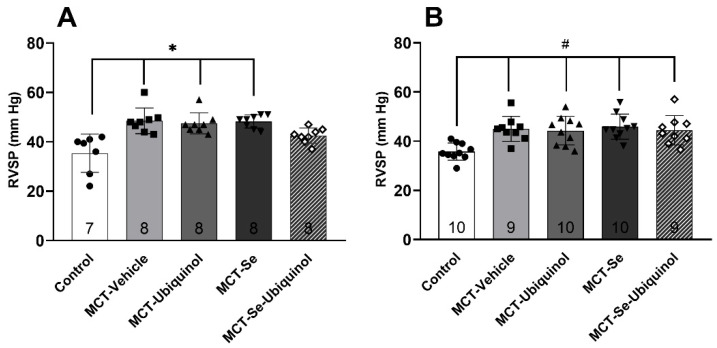
Right ventricle systolic pressure (RVSP) in male rats (**A**) and female rats (**B**) on day 21 of the experiment (mean ± SD). (**A**) Male rats * Control vs. MCT-Vehicle, MCT-Ubiquinol, MCT-Se, *p* < 0.01, (**B**) Female rats ^#^ Control vs. MCT-Vehicle, MCT-Ubiquinol, MCT-Se, MCT-Se-Ubiquinol, *p* < 0.05, one-way ANOVA.

**Figure 4 antioxidants-11-00549-f004:**
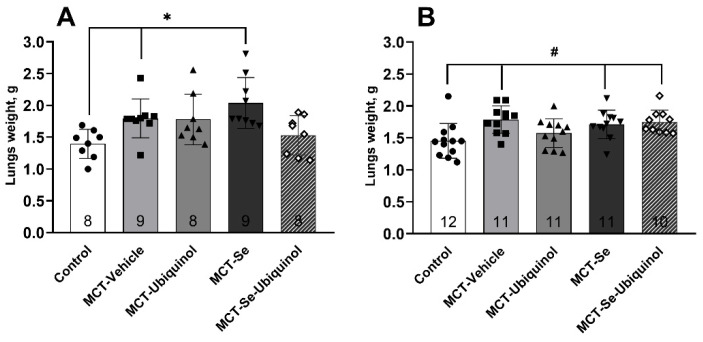
Lung mass in male rats (**A**) and female rats (**B**) on day 21 of the experiment (mean ± SD). Male rats * Control vs. MCT-Vehicle, MCT-Se, *p* < 0.05, female rats # Control vs. MCT-Vehicle, MCT-Se, MCT-Se-Ubiquinol, *p* < 0.05, one-way ANOVA.

**Figure 5 antioxidants-11-00549-f005:**
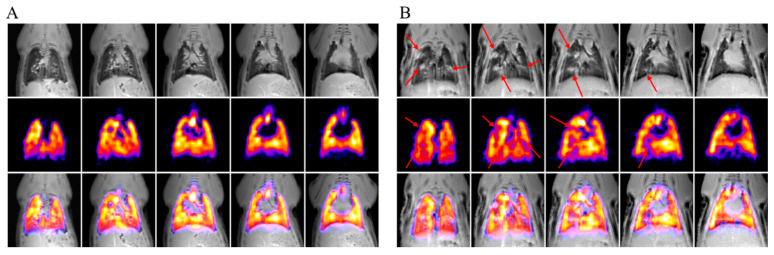
MRI of two rats without (**A**) and with (**B**) lung fibrosis: selected slices of ^1^H (top rows) and ^19^F (middle rows) MRI in coronal projection, as well as their overlaying (bottom rows). The red arrows point to fibrotic changes in the lungs.

**Table 1 antioxidants-11-00549-t001:** Glutathione peroxidase activity.

Group	Control (*n* = 6)	MCT-Vehicle (*n* = 6)	MCT-Ubiquinol (*n* = 6)	MCT-Se (*n* = 6)	MCT-Se-Ubiquinol (*n* = 6)
μmol of NADPH/min × V sample					
**Males**	7.021 ± 0.826	8.079 ± 0.266 *,^#^	6.611 ± 0.593	6.348 ± 0.569	7.172 ± 0.364
**Females**	10.880 ± 1.792 ^$^	9.365 ± 1.346	9.420 ± 0.441 ^$^	9.764 ± 1.418 ^$^	9.637 ± 0.411 ^$^

Male rats * MCT-Vehicle vs. Control, *p* < 0.05; ^#^ MCT-Vehicle vs. MCT-Ubiquinol, MCT-Se, *p* < 0.01, one-way ANOVA; ^$^ females vs. males, *p* < 0.01, two-way ANOVA.

**Table 2 antioxidants-11-00549-t002:** Index of hypertrophy (RV mass/body weight × 1000), (mean ± SD).

Group	Control	MCT-Vehicle	MCT-Ubiquinol	MCT-Se	MCT-Se-Ubiquinol
**Males**	0.611 ± 0.057 (*n* = 8)	0.877 ± 0.328 * (*n* = 9)	0.740 ± 0.115 (*n* = 8)	0.796 ± 0.171 ^&^ (*n* = 9)	0.751 ± 0.094 (*n* = 8)
**Females**	0.636 ± 0.083 (*n* = 12)	0.828 ± 0.248 ^#^ (*n* = 11)	0.754 ± 0.083 (*n* = 11)	0.792 ± 0.148 ^$^ (*n* = 11)	0.733 ± 0.113 (*n* = 10)

Male rats * MCT-Vehicle, ^&^ MCT-Se vs. Control, *p* < 0.05; female rats ^#^ MCT-Vehicle, ^$^ MCT-Se vs. Control, *p* < 0.05, one-way ANOVA.

**Table 3 antioxidants-11-00549-t003:** The total number of animals with an altered lung structure and ventilation disorder in female rats.

Group	^1^H MRI Signs of Fibrotic Changes	^19^F MRI Disruption in Ventilation
Control (*n* = 4)	0	0
MCT-Vehicle (*n* = 4)	4 *	4 ^$^
MCT-Ubiquinol (*n* = 4)	2	3 ^&^
MCT-Se (*n* = 4)	4 *	4 ^$^
MCT-Se-Ubiquinol (*n* = 4)	3 ^#^	4 ^$^

* MCT-Vehicle, MCT-Se vs. Control, *p* < 0.01, **^#^** MCT-Se-Ubiquinol vs. Control, *p* < 0.05; ^$^ MCT-Vehicle, MCT-Se, MCT-Se-Ubiquinol vs. Control, *p* < 0.001, ^&^ MCT-Ubiquinol vs. Control, *p* < 0.01, Kruskal-Wallis H test.

## Data Availability

The data presented in this study are available on request from the corresponding author. The data are not publicly available due to a large amount of digital information requiring special equipment.

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
