# Peer review of "Sex-Dependent Protective Effect of Combined Application of Solubilized Ubiquinol and Selenium on Monocrotaline-Induced Pulmonary Hypertension in Wistar Rats"

_antioxidants, 2022, doi:10.3390/antiox11030549_

Round 1
Reviewer 1 Report
I have some suggestions to improve the quality of this paper.
- Also taking into consideration the detailed description of methods, the applied statistics shoud be mentioned in this section.
- The last sentence of the abstract is not clear
- line 61, Introduction. Regarding the rationale of the study it should be mentioned that thioredoxine reductase is also a selenium-dependent ubiquinone reductase (see Ling Xia et al, JBC 2003, Nordman T, Biofactors 2003), which regenerates ubiquinol.
- Line 110: what is solubilized ubiquinol? A brief description of the ingrediens should be given.
- Line 119: Female rats underwent lung MRI; what about male rats?
- Fig 1: unify selenium dose with what is described in the text. In line 115 selenium 100 mg is mentioned. There should be no discrepancy.
- Results, line 257, 30 mg/kg
- line 281, RVSP registration: what is "average rate" of RVSP?
- Table 3: how was the statistics performed?
- Line 327: ....displayed more rarely... Was it statistically different?
- Discussion line 338 and following. Ameliorating effect of Coenzyme Q10 on endothelial disfunction in humans affected by ischemic heart disease is described in the literature (see Belardinelli et al 2006, Tiano et al 2007). Those results should be mentioned in the discussion, since they are in agreement with the findings of the present study.
Author Response
Point 1: Also taking into consideration the detailed description of methods, the applied statistics shoud be mentioned in this section.
Response 1: We have included information about statistical analysis in the "materials and methods" section (line 269).
Point 2: The last sentence of the abstract is not clear
Response 2: The last sentence of the abstract has been corrected (line 32)
Point 3: Line 61, Introduction. Regarding the rationale of the study it should be mentioned that thioredoxine reductase is also a selenium-dependent ubiquinone reductase (see Ling Xia et al, JBC 2003, Nordman T, Biofactors 2003), which regenerates ubiquinol.
Response 3: Both articles have been included in the main text and the list of references (line 65)
Point 4: Line 110: what is solubilized ubiquinol? A brief description of the ingrediens should be given.
Response 4: We have added a brief description about the consumption of solubilized ubiquinol. Composition (1 mL): ubiquinol 10 mg, macrogol glycerol ricyl oleate 80 mg, polysorbate 40 mg, ascorbic acid 1 mg, sodium EDTA 0.5 mg, sodium chloride 0.9% ad 1 mL); the MCT-Vehicle group receiving vehicle for the ubiquinol substance (the same composition, but without ubiquinol i.e. it was the control group to ubiquinol administration) (line 127).
Point 5: Line 119: Female rats underwent lung MRI; what about male rats?
Response 5: Since the MCT-Ubiquinol and MCT Se-Ubiquinol groups of male rats at the first part of the experiment did not reveal any significant increase in lung mass, the subsequent experiment on female rats involved MRI of lungs on day 20 to detect fibrotic and unventilated regions to check the assumption of the ubiquinol influence.
Point 6: Fig 1: unify selenium dose with what is described in the text. In line 115 selenium 100 mg is mentioned. There should be no discrepancy.
Response 6: The discrepancy has been corrected (line 123).
Point 7: Results, line 257, 30 mg/kg
Response 7: The mistake has been corrected (line 282).
Point 8: line 281, RVSP registration: what is "average rate" of RVSP?
Response 8: Average rate=mean rate. RVSP registration displayed that all the groups with subcutaneous MCT administration, except the MCT-Se-Ubiquinol male group (fig.A), exhibited а significant increase in this parameter in comparison with the groups of healthy animals (p<0.05).
Point 9: Table 3: how was the statistics performed?
Response 9: The statistics in table 3 was performed as follows: animals with signs of pulmonary fibrosis/ ventilatory impairment on MRI imaging were marked with number 1, and those without number 0, were further analyzed using Kruskal-Wallis H test.
Point 10: Line 327: ....displayed more rarely... Was it statistically different?
Response 10: The sentence has been corrected (line 352).
Point 11: Discussion line 338 and following. Ameliorating effect of Coenzyme Q10 on endothelial disfunction in humans affected by ischemic heart disease is described in the literature (see Belardinelli et al 2006, Tiano et al 2007). Those results should be mentioned in the discussion, since they are in agreement with the findings of the present study.
Response 11: Both articles (Belardinelli et al 2006, Tiano et al 2007) were included in the mail text and references.
Reviewer 2 Report
This paper evaluated the effects of ubiquinol and selenium administration in rats model of pulmonary hypertension. The results are scientifically valid, and the manuscript is well-written, however the authors need to specify following items:
In full document insert the concept in full before the acronym for the first time it is used. In the title avoid acronym for “MCT” for “Monocrotaline”. Check the abstract and specify full name before acronym (see lines 16, 22, 24 and 25). Check full paper.
Line 22: the sentence starts with “On day 22”… insert lacking sentence.
Line 64: N-terminal pro-brain natriuretic peptide (NT-proBNP) levels.
Line 106: specify how many animals per group.
Line 115: specify dosage of selenium administration.
Line 298: Table 2 specify notes for each symbol *,&,#,$
Line 358: specify the reason for choosing the dosage of oral selenium administration.
Line 450: please, provide explanation for the sentence “Selenium supplement probably neutralized the ubiquinol action in the female”. Did the authors consider an eventual lack of effect in selenium group due to an insufficient selenium dosage? Please discuss about it.
Author Response
Point 1: In full document insert the concept in full before the acronym for the first time it is used. In the title avoid acronym for “MCT” for “Monocrotaline”. Check the abstract and specify full name before acronym (see lines 16, 22, 24 and 25). Check full paper.
Response 1: We have traced how the acronyms are marked for the first time and have given them a transcription if it was necessary
Point 2: Line 22: the sentence starts with “On day 22”… insert lacking sentence.
Response 2: Line 22. The sentence is completed.
Point 3: Line 64: N-terminal pro-brain natriuretic peptide (NT-proBNP) levels.
Response 3: Line 70. The acronym is transcripted.
Point 4: Line 106: specify how many animals per group.
Response 4: We have added the information about how many animals per group were initially (line 116) and then added n in certain results if it was necessary
Point 5: Line 115: specify dosage of selenium administration.
Response 5: A wide range of oral doses of selenium is used in experiments on animals; however, if the physiological requirement is exceeded, selenium can become toxic. Due to the fact that small quantities, calculated for an average rat weight of 200-300 g, are very easy to overdose, a dosage of 10 µg/kg was chosen (line 136).
Point 6: Line 298: Table 2 specify notes for each symbol *,&,#,$
Response 6: The symbols are transcripted (line 322).
Point 7: Line 358: specify the reason for choosing the dosage of oral selenium administration.
Response 7: We have added this information in the end of «materials and methods» (line 136).
Point 8: Line 450: please, provide explanation for the sentence “Selenium supplement probably neutralized the ubiquinol action in the female”. Did the authors consider an eventual lack of effect in selenium group due to an insufficient selenium dosage? Please discuss about it.
Response 8: Yes, we consider that the dose of 10 mcg might be not enough to have an effect on the expression of GPx and also selenium incorporation depends on the estrogen cycle (line 435).
Reviewer 3 Report
The manuscript submitted by Kuropatkina and colleagues reveals sex-dependent protective effect of combined application of ubiquinol and selenium on a model of pulmonary hypertension (PH) in Wistar Rats. To evaluate the effect of these substances on MCT-induced PH development, they asses the antioxidant status, RVSP and RV hypertrophy and they use MRI to monitor lung status.
No differences were observed in RVSP in females with Se-Ub administration, whereas males show a decrease in RVSP when Se and Ub were administered. Do the authors have any hypothesis to explain the observed sex-differences?
Did the authors perform any determination of NO levels?
The authors propose that Se could neutralize Ub action in the female group and propose an association with the expression of selenoprotein P. Did the authors perform any assay to support this assumption in males and females?
They also show reduced fibrosis in the groups receiving ubiquinol. Did the authors determine any oxidative stress or inflammation marker?
Could the authors clarify statistic in figure legend from Table 2?
Author Response
Point 1: No differences were observed in RVSP in females with Se-Ub administration, whereas males show a decrease in RVSP when Se and Ub were administered. Do the authors have any hypothesis to explain the observed sex-differences?
Response 1: Yes, we do. We assume that lack of effect might be because of estrogen cycle that influences selenium incorporation (line 435)
Point 2: Did the authors perform any determination of NO levels?
Response 2: No, we didn’t. Previously, in a similar experiment on males (selenium hasn’t been studied there), we determined the concentration of nitric oxide metabolites in the daily urine, and obtained a trend towards an increase in NO2--ions in the MCT-Ubiquinol group, but the standard deviation was too large, that is why these data have not been published anywhere.
Point 3: The authors propose that Se could neutralize Ub action in the female group and propose an association with the expression of selenoprotein P. Did the authors perform any assay to support this assumption in males and females?
Response 3: Unfortunately, we didn’t.
Point 4: They also show reduced fibrosis in the groups receiving ubiquinol. Did the authors determine any oxidative stress or inflammation marker?
Response 4: In this work, we did not analyze any markers of oxidative stress or inflammation. However, previously we evaluated the expression level of pro-inflammatory marker micro-RNA-34a in right ventricular heart tissue on a more severe model of 4-week MCT-induced PH (selenium hasn’t been studied there). We showed significantly lower expression level in MCT-ubiquinol group compared with MCT-Vehicle group.
DOI 10.3897/rrpharmacology.7.67291
Point 5: Could the authors clarify statistic in figure legend from Table 2?
Response 5: The legend from table 2 has been corrected.